EMBO
Molecular Medicine

# Cobalt protoporphyrin IX increases endogenous G-CSF and mobilizes HSC and granulocytes to the blood

Agata Szade[1,*], Krzysztof Szade[1], Witold N Nowak[1], Karolina Bukowska-Strakova[1,2], Lucie Muchova[3], Monika Gońka[1], Monika Żukowska[1], Maciej Cieśla[1,†], Neli Kachamakova-Trojanowska[1,4], Marzena Rams-Baron[5,6], Alicja Ratuszna[5,6], Józef Dulak[1,4] & Alicja Józkowicz[1]

## Abstract

Granulocyte colony-stimulating factor (G-CSF) is used in clinical practice to mobilize cells from the bone marrow to the blood; however, it is not always effective. We show that cobalt protoporphyrin IX (CoPP) increases plasma concentrations of G-CSF, IL-6, and MCP-1 in mice, triggering the mobilization of granulocytes and hematopoietic stem and progenitor cells (HSPC). Compared with recombinant G-CSF, CoPP mobilizes higher number of HSPC and mature granulocytes. In contrast to G-CSF, CoPP does not increase the number of circulating T cells. Transplantation of CoPP-mobilized peripheral blood mononuclear cells (PBMC) results in higher chimerism and faster hematopoietic reconstitution than transplantation of PBMC mobilized by G-CSF. Although CoPP is used to activate Nrf2/HO-1 axis, the observed effects are Nrf2/HO-1 independent. Concluding, CoPP increases expression of mobilization-related cytokines and has superior mobilizing efficiency compared with recombinant G-CSF. This observation could lead to the development of new strategies for the treatment of neutropenia and HSPC transplantation.

**Keywords** CoPP; granulocyte colony-stimulating factor; hematopoietic stem and progenitor cells; HO-1; mobilization

**Subject Categories** Haematology; Stem Cells & Regenerative Medicine

## Introduction

Porphyrins are macrocyclic compounds essential for plants, bacteria, and animals, found in molecules such as chlorophylls and cytochromes (Chandra *et al*, 2000). Porphyrins form complexes with metals to generate metalloporphyrins. Often the bound metal ions determine the unique properties of the metalloporphyrins. For example, only Fe-protoporphyrin IX (heme) is a substrate for heme oxygenase-1 (HO-1), but other protoporphyrins such as tin protoporphyrin IX (SnPP) can inhibit HO-1 enzymatic activity (Schulz *et al*, 2012) (Fig 1A and B). Cobalt protoporphyrin (CoPP) is the inducer of Nrf-2/HO-1 axis, both *in vitro* and *in vivo*. Thus, CoPP is considered as a potential inducer of HO-1 where it may have therapeutic advantages (Shan *et al*, 2006).

Heme oxygenase-1 is an enzyme which degrades heme into carbon monoxide (CO), iron ions, and biliverdin (Fig 1B), which is subsequently reduced to bilirubin (Tenhunen *et al*, 1968). The idea of activation of HO-1 for therapeutic purposes is based on its broad anti-inflammatory properties (Ryter *et al*, 2006). HO-1 also influences the maturation and activity of myeloid cells. Specific deletion of HO-1 in myeloid lineage (Lyz-Cre:Hmox1$^{fl/fl}$) partially blocks differentiation of myeloid progenitors toward macrophages (Wegiel *et al*, 2014). We showed that lack of HO-1 also affects granulopoiesis (Bukowska-Strakova *et al*, 2017). HO-1$^{-/-}$ mice have more granulocytes in the peripheral blood (PB), what is connected with increased myelocyte proliferation in the bone marrow (BM). Moreover, CoPP may inhibit the maturation of dendritic cells (Chauveau *et al*, 2005), but this effect seems to be HO-1 independent (Mashreghi *et al*, 2008).

1 Department of Medical Biotechnology, Faculty of Biochemistry, Biophysics and Biotechnology, Jagiellonian University, Krakow, Poland
2 Department of Clinical Immunology and Transplantology, Institute of Pediatrics, Jagiellonian University Medical College, Krakow, Poland
3 Fourth Department of Internal Medicine and Institute of Medical Biochemistry and Laboratory Medicine, First Faculty of Medicine, Charles University in Prague, Prague, Czech Republic
4 Malopolska Centre of Biotechnology, Jagiellonian University, Krakow, Poland
5 A. Chelkowski Institute of Physics, University of Silesia, Chorzow, Poland
6 Silesian Center for Education and Interdisciplinary Research, Chorzow, Poland
*Corresponding author: Tel: +48 12 6646024; E-mail: agata.szade@uj.edu.pl
†Present address: Division of Molecular Hematology, Lund University, Lund, Sweden

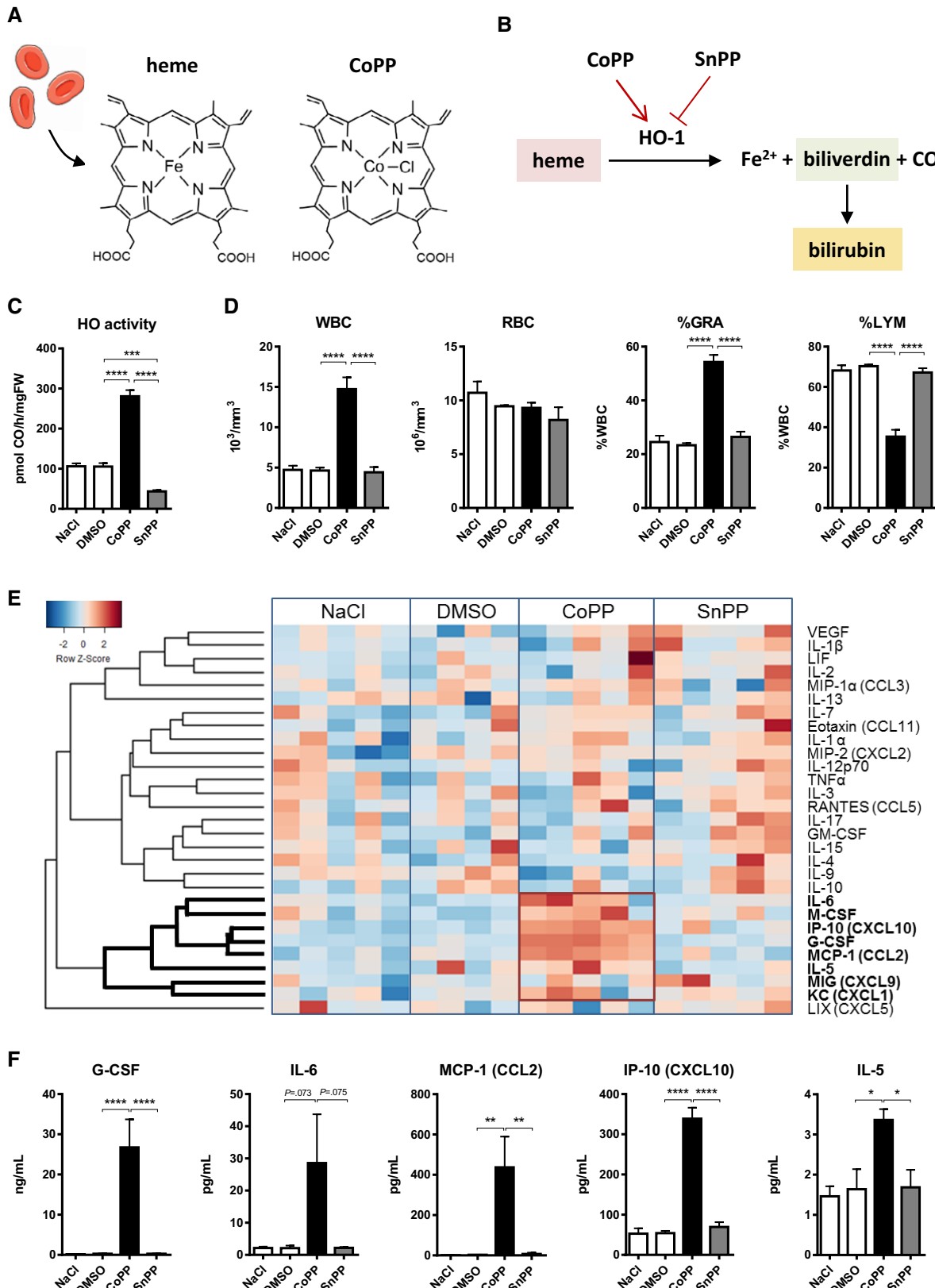

**Figure 1.**

**Figure 1.  CoPP increases number of granulocytes and upregulates a set of cytokines.**

C3H mice were injected with CoPP, SnPP, or solvent controls (NaCl, DMSO) each second day for 5 days. Samples were collected 24 h after the last injection.

A  Chemical structures of heme (HO-1 substrate) and CoPP (HO-1 *in vivo* inducer).
B  Heme degradation reaction catalyzed by HO-1.
C  Heme oxygenase activity is increased by CoPP and decreased by SnPP in the liver as measured by gas chromatography.
D  Total leukocyte and red blood cell count in PB. CoPP increases the number of major WBC types. Granulocyte and monocyte percentages are increasing, whereas lymphocyte percentage is decreasing among PB leukocytes after CoPP.
E  Heatplot of the cytokine and growth factor concentrations in plasma measured by Luminex assay. CoPP increases the concentrations of a group of cytokines (red box).
F  Selected cytokine and growth factor concentrations in plasma measured by Luminex assay. CoPP increases concentrations of G-CSF, MCP-1, IP-10, IL-6, and IL-5.

Data information: Results are shown as mean + SEM, one-way ANOVA with Bonferroni post-test, $n$ = 5 mice per group. *$P \leq 0.05$; **$P \leq 0.01$; ***$P \leq 0.001$; ****$P \leq 0.0001$.

Given this rationale, we sought to study how pharmacological induction of HO-1 by CoPP can influence the function of myeloid lineage. Unexpectedly, we observed that CoPP efficiently mobilizes granulocytes as well as hematopoietic stem and progenitor cells to the PB in mice. Because of constant need to improve clinical mobilization strategies, we investigated the effects induced by CoPP as a potential new approach to meet this requirement.

During mobilization, hematopoiesis is enhanced, and large numbers of cells are released from the BM to the blood, including stem and immature hematopoietic cells (Lapidot & Petit, 2002). Mechanism of mobilization is complex and involves several cell populations and cytokine pathways (Duhrsen *et al*, 1988; Lapidot & Petit, 2002; Tay *et al*, 2017), with G-CSF (granulocyte colony-stimulating factor) as one of the best characterized mobilizing factors (Souza *et al*, 1986; Lapid *et al*, 2008). G-CSF acts on the BM myeloid progenitors, driving their proliferation and differentiation toward granulocytes (Metcalf & Nicola, 1983). Apart from G-CSF, many other agents, such as stromal cell-derived factor 1α (Hattori *et al*, 2001; Devine *et al*, 2008), stem cell factor (Andrews *et al*, 1992), interleukin 6 (IL-6) (Pojda & Tsuboi, 1990), IL-8 (Laterveer *et al*, 1995), Groβ (Pelus & Fukuda, 2006; Fukuda *et al*, 2007), or granulocyte-macrophage colony-stimulating factor (GM-CSF) (Gianni *et al*, 1989) can act as mobilizing factors (reviewed in (Lapid *et al*, 2008)). Recently, Hoggatt *et al* (2018) reported rapid mobilization of highly engrafting stem cells with a single injection of Groβ and AMD3100 combination.

Current progress in basic science concerning cell mobilization has already been successfully translated into clinical practice (Bronchud *et al*, 1987; Sheridan *et al*, 1992). Pharmacological mobilization is of outstanding importance for the prevention or treatment of neutropenia (Kelly & Wheatley, 2009; Lyman *et al*, 2010) and for the transplantation of hematopoietic stem cells (HSC). The success of BM transplantation depends on the collection of sufficient number of HSC (Kondo *et al*, 2008). Nowadays, the source of transplantable HSC is not necessarily BM itself, but rather the PB after mobilization of HSC into the circulation (Cashen *et al*, 2007; To *et al*, 2011).

Recombinant human G-CSF is widely used for mobilization purposes (Mehta *et al*, 2015). Despite improvements in the treatment protocols, in some patients application of G-CSF is inefficient. Among healthy donors, G-CSF mobilization fails in 5–30%, but in high-risk patients, the failure rate reaches even up to 60% (Ferraro *et al*, 2011; To *et al*, 2011). Patients who fail to mobilize HSC in response to G-CSF might be treated additionally with plerixafor (To *et al*, 2011); however, the price of this drug is sometimes a major obstacle—single dose costs several thousand dollars. Therefore, to

improve the treatment efficiency, we need new agents with additional activities. These can include modulating of extracellular matrix (Saez *et al*, 2014), phosphorylation of signaling proteins (Wang *et al*, 2016), inhibition of proteasome (Ghobadi *et al*, 2014), or induction of endogenous G-CSF (Hoggatt & Pelus, 2014). Here, we describe the previously unknown mobilizing properties of CoPP and compare its efficiency to the standard mobilizing factor, G-CSF.

# Results

## CoPP treatment increases leukocyte numbers in the blood

CoPP and SnPP are commonly used as activator and inhibitor of HO-1, respectively (Fig 1B) (Ryter *et al*, 2006). As expected, administration of CoPP to C3H mice resulted in a 2.6-fold increase in HO-1 activity in the liver, whereas SnPP decreased HO-1 activity by 2.4 times (Fig 1C). Along with increased HO-1 activity, mice treated with CoPP had increased absolute number of all types of leukocytes (WBC) in the blood, with a visible shift toward myeloid lineage (Fig 1D). Erythrocyte parameters were unaffected by CoPP treatment (Fig 1D).

To examine whether the observed leukocytosis was linked with changes in cytokine profile in plasma, we performed Luminex screen on 32 cytokines (Fig 1E). CoPP increased concentrations of the set of cytokines (Fig 1E,F) that includes IL-6, monocyte chemoattractant protein 1 (MCP-1, CCL2), interferon γ-induced protein 10 (IP-10, CXCL10), IL-5, and to the greatest extent G-CSF (Fig 1F). However, CoPP did not increase the other CSFs—M-CSF, GM-CSF, and IL-3 (Appendix Fig S1). SnPP, HO-1 inhibitor, did not influence the complete blood cell count or any of the analyzed cytokines (Fig 1D–F, Appendix Fig S1).

## Both CoPP and G-CSF mobilize myeloid cells but differ in T-cell mobilization and upregulation of cytokines in plasma

As the treatment of mice with CoPP increased G-CSF concentration in plasma (Fig 1F), and G-CSF is known clinical mobilizing agent, we directly compared the effects of G-CSF and CoPP administration. We injected the mice with G-CSF or CoPP once a day for 5 days and analyzed myeloid and hematopoietic stem/progenitor cells in PB and BM by flow cytometry.

Both G-CSF and CoPP increased the number of CD45$^+$ cells in PB (Fig 2A). Although both in G-CSF- and CoPP-treated mice the highest increase was observed for granulocytes, G-CSF mobilized 1.6 times more total granulocytes than CoPP. The numbers of

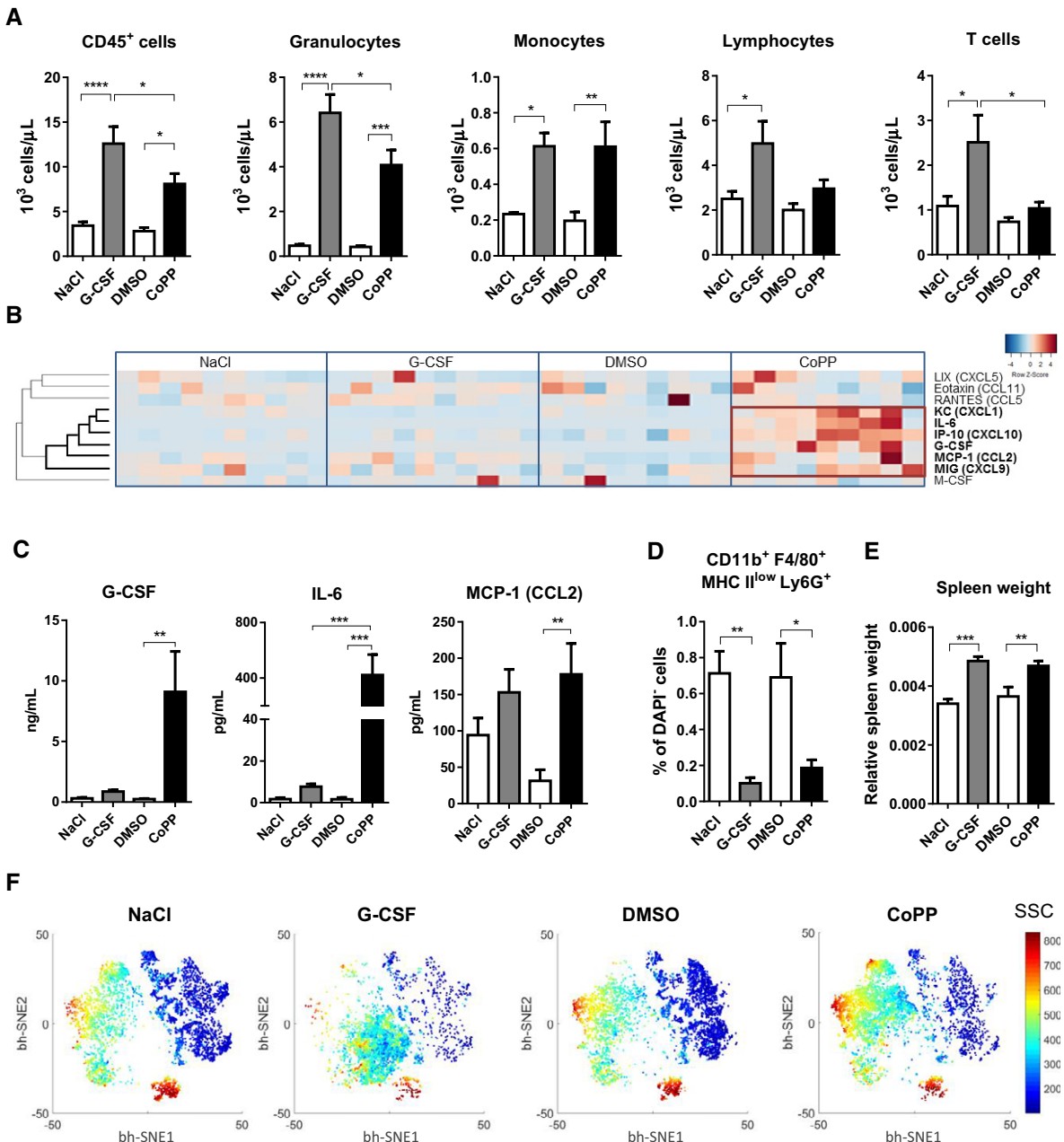

**Figure 2. Both G-CSF and CoPP mobilize myeloid cells, but only G-CSF mobilizes T cells.**

C57BL6xFVB mice were injected with G-CSF, CoPP, or solvent controls (NaCl, DMSO) daily for 5 days. Samples were collected 6 h after the last injection.

A    Cell numbers of main leukocyte populations in PB measured by flow cytometry. Both G-CSF and CoPP increase numbers of CD45+ cells in blood. G-CSF and CoPP similarly increase monocyte numbers, but the increase in granulocytes is higher after G-CSF. Lymphocytes, including T cells, are increased only by G-CSF and not by CoPP.

B    Cytokine and growth factor concentrations in plasma measured by Luminex assay. CoPP induces a group of cytokines (red box) which are not induced by G-CSF.

C    Plasma concentrations of selected cytokines and growth factors. CoPP highly increases concentration of endogenous G-CSF and IL-6 that are not increased by G-CSF.

D, E    Treatment with G-CSF and CoPP (D) decreases percentage of Ly6G+ macrophages in the BM and (E) increases relative spleen weight.

F    viSNE maps of CD11b+ CD11c− blood cells, colored by SSC value. Three independent experiments were performed, and the results were pooled.

Data information: Results are shown as mean + SEM, one-way ANOVA with Bonferroni post-test, $n = 7$ mice per group. *$P \leq 0.05$; **$P \leq 0.01$; ***$P \leq 0.001$; ****$P \leq 0.0001$.

monocytes were similarly increased after G-CSF and CoPP. Importantly, only G-CSF treatment led to the increase in PB lymphocytes, mainly T cells (Fig 2A).

Next, we compared cytokine concentrations in plasma using Luminex assay (Fig 2B). Consistently with the previous experiment, CoPP induced high levels of endogenous G-CSF, IL-6, and MCP-1

(Fig 2C). CoPP treatment also elevated KC (keratinocyte-derived cytokine, CXCL1), IP-10, and MIG (monokine induced by interferon γ, CXCL9). In contrast, G-CSF did not significantly induce any of these cytokines (Fig 2B).

Analyzing the other features related to cell mobilization (Platz-becker *et al*, 2001; Winkler *et al*, 2010), we found that mice treated with both G-CSF and CoPP had decreased numbers of CD11b$^+$ F4/80$^+$ MHC II$^{low}$ Ly6G$^+$ macrophages in the BM (Fig 2D) and enlarged spleens (Fig 2E).

Concluding, both CoPP and G-CSF mobilize efficiently myeloid cells, but result in different mobilization of lymphocytes. CoPP induces several cytokines together with endogenous G-CSF that are not elevated during mobilization with recombinant G-CSF.

## CoPP mobilizes granulocytes with mature phenotype

Although both G-CSF and CoPP efficiently mobilize granulocytes, we observed that these cells have a distinct phenotype. Multiparameter analysis revealed that myeloid cells from the mice treated with CoPP are more similar to the cells from the control mice than to the cells from the G-CSF-treated mice (Fig 2F).

viSNE maps showed increased density of the granular cells in the CoPP group compared with the control. We also observed a population of cells with intermediate granularity in mice treated with G-CSF that was not visible in the control mice (Fig 2F). Accordingly, 2D flow cytometry plots confirm that CoPP mobilizes granulocytes that are more granular (higher SSC parameter) and have higher expression of Ly6G comparing to granulocytes mobilized by G-CSF (Fig 3A). Granulocytes mobilized by CoPP phenotypically resemble the mature granulocytes in control mice, while these mobilized by G-CSF show immature phenotype, with lower granularity and Ly6G expression, typical for early differentiation stages in BM. Altogether, although mice treated with G-CSF had a higher total number of granulocytes in the blood, these were mainly immature (Ly6G$^{mid}$ SSC$^{mid}$), whereas CoPP treatment increased the number of mature cells (Ly6G$^{hi}$ SSC$^{hi}$; Fig 3B).

Next, we analyzed the composition of myeloid cell populations in the BM. Both G-CSF and CoPP treatment increased the frequency of granulocytes with the immature phenotype, (CD11b$^+$ CD11c$^-$ Ly6C$^{low}$ Ly6G$^+$ and CD11b$^+$ CD11c$^-$ Ly6C$^{low}$ SSC$^{med}$ Ly6G$^{med}$), but the increase after G-CSF was more pronounced. Percentage of granulocytes with the mature phenotype (CD11b$^+$ CD11c$^-$ Ly6C$^{low}$ SSC$^{hi}$ Ly6G$^{hi}$) was decreased after G-CSF, but not affected by CoPP (Fig 3C).

To verify the functional properties of mobilized granulocytes, we checked the production of reactive oxygen species (ROS). For this purpose, G-CSF- and CoPP-mobilized or control PB was incubated with *N*-formylmethionyl-leucyl-phenylalanine (fMLP), phorbol 12-myristate 13-acetate (PMA) or opsonized *Escherichia coli* and subjected to rhodamine 123 (DHR 123) staining (Fig 3D). There was a higher percentage of ROS-producing cells in the blood of G-CSF- or CoPP-treated mice than in control mice, after stimulation with *E. coli*, with similar tendency after PMA treatment. We did not observe any differences between groups after stimulation with fMLP. Interestingly, mobilized granulocytes with immature phenotype seemed to be able to produce ROS after stimulation with PMA (CoPP- and G-CSF-mobilized cells) and *E. coli* (G-CSF-mobilized cells, Appendix Fig S2).

Concluding, granulocytes mobilized either by G-CSF or by CoPP are at least as functional as the steady-state granulocytes in tested conditions.

## CoPP mobilizes more HSPC than G-CSF

As G-CSF also mobilizes hematopoietic stem and progenitor cells (Lapidot & Petit, 2002), we analyzed HSPC populations in the blood and BM of mice treated with G-CSF and CoPP. CoPP increased percentage and number of total HSPC pool defined as c-Kit$^+$ Lin$^-$ Sca-1$^+$ (KLS) cells in the blood (Appendix Fig S3A). In animals treated with G-CSF, the increase in KLS cells was visible, although not statistically significant when all four groups were compared together (one-way ANOVA with Bonferroni post-test). We further characterized mobilized KLS pool using CD34 and SLAM markers: CD48 and CD150, which enable to define HSC (KLS CD48$^-$CD150$^+$), MPP (multipotent progenitors, KLS CD48$^-$CD150$^-$), and HPC (hematopoietic progenitors, KLS CD48$^+$CD150$^-$ and KLS CD48$^+$CD150$^+$) populations (Fig 4A) (Oguro *et al*, 2013). CoPP treatment mobilized more HSC, MPP, and HPC than the treatment with G-CSF (Fig 4B). Of note, only small proportion of KLS cells mobilized by G-CSF and CoPP were CD34 negative (Appendix Fig S3B), what is consistent with the previous observation, that G-CSF-mobilized HSC (in contrast to steady-state HSC) are CD34$^+$ (Tajima *et al*, 2000). Next, we compared how G-CSF and CoPP affect HSPC in BM. Only CoPP treatment significantly increased the percentage of KLS cells in BM (Appendix Fig S3A), but this increase was restricted to more differentiated HPC fraction and was not observed among LT-HSC (long-term HSC; KLS CD48$^-$CD150$^+$CD34$^-$) and MPP populations (Fig 4C).

Cobalt protoporphyrin mobilization resulted in higher number of more committed progenitors (Fig 4D) in PB: granulocyte-macrophage progenitors (GMP) and megakaryocyte–erythroid progenitors (MEP). The increase in erythroid progenitors (EP) number was similar in both groups. In contrast, both G-CSF and CoPP decreased percentage of committed progenitors c-Kit$^+$Lin$^-$Sca-1$^-$ (KLS$^-$) in BM; however, the decrease after CoPP was less pronounced (Fig 4E). Further characterization of committed progenitors with CD34, CD48, and CD150 markers showed that MEP and EP are similarly decreased after G-CSF and CoPP. Only G-CSF, but not CoPP decreased percentage of GMP in BM, that did not change after CoPP treatment (Fig 4E). Together with the observation that there was a higher number of mobilized GMP by CoPP, it might suggest that CoPP or its downstream effectors affect GMP proliferation.

## CoPP mobilizes functional HSPC

Cobalt protoporphyrin increased the number of HSPC in the PB more efficiently than G-CSF. To verify whether CoPP-mobilized HSPC are functional, we transplanted the mobilized PB mononuclear cells (PBMC) and checked their hematopoietic reconstitution potential.

We treated the green fluorescent protein (GFP)-expressing mice (C57BL/6-Tg(UBC-GFP)30Scha/J) with CoPP, G-CSF, or NaCl (Fig EV2). At the fifth day of the treatment, we collected the blood and transplanted 5 × 10$^6$ PBMC to the lethally irradiated GFP$^-$ recipient mice, together with 10$^5$ GFP$^-$ BM-derived competitor cells. Two independent experiments were performed.

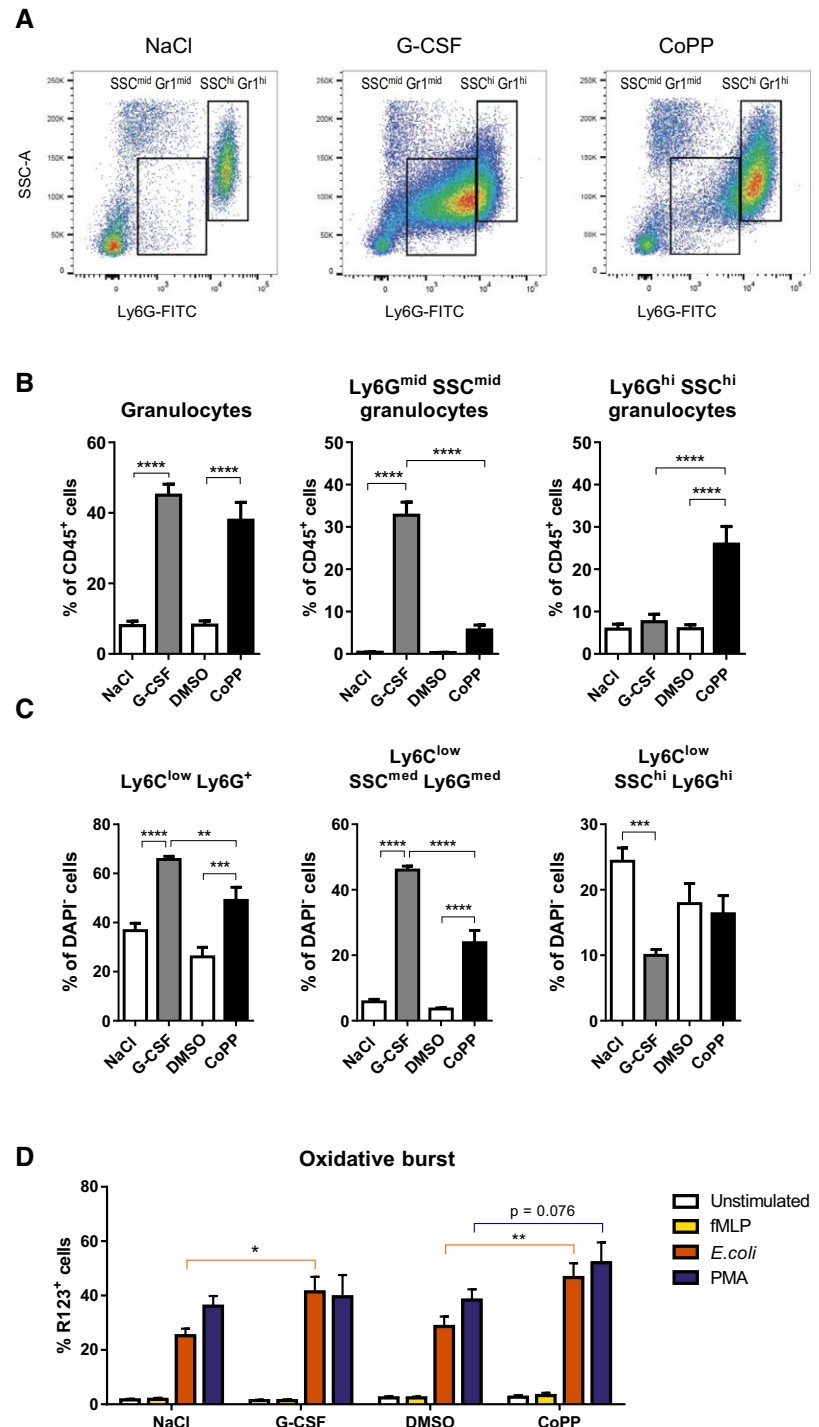

**Figure 3. CoPP and G-CSF mobilize granulocytes with different phenotype.**

C57BL/6 mice were treated with G-CSF, CoPP, or solvent controls (NaCl, DMSO) for five consecutive days. Samples were collected 6 h after the last injection.

A   Representative flow cytometry plots show higher granularity (SSC) and Ly6G expression in CoPP-mobilized granulocytes than in G-CSF mobilized granulocytes.

B, C   Relative abundance of different granulocyte phenotypes in blood (B) or BM (C) of mice treated with G-CSF and CoPP. Mice treated with CoPP have higher proportion of granulocytes with mature phenotype, than mice treated with G-CSF.

D   The percentage of cells isolated from C57BL/6xFVB mice treated with G-CSF or CoPP that are producing reactive oxygen species after incubation with indicated stimuli.

Data information: Results are shown as mean + SEM, one-way ANOVA with Bonferroni post-test, $n = 7$ mice per group (B, C) or two-way ANOVA with Bonferroni post-test, $n = 6$ mice per group (D). *$P \leq 0.05$; **$P \leq 0.01$; ***$P \leq 0.001$; ****$P \leq 0.0001$.

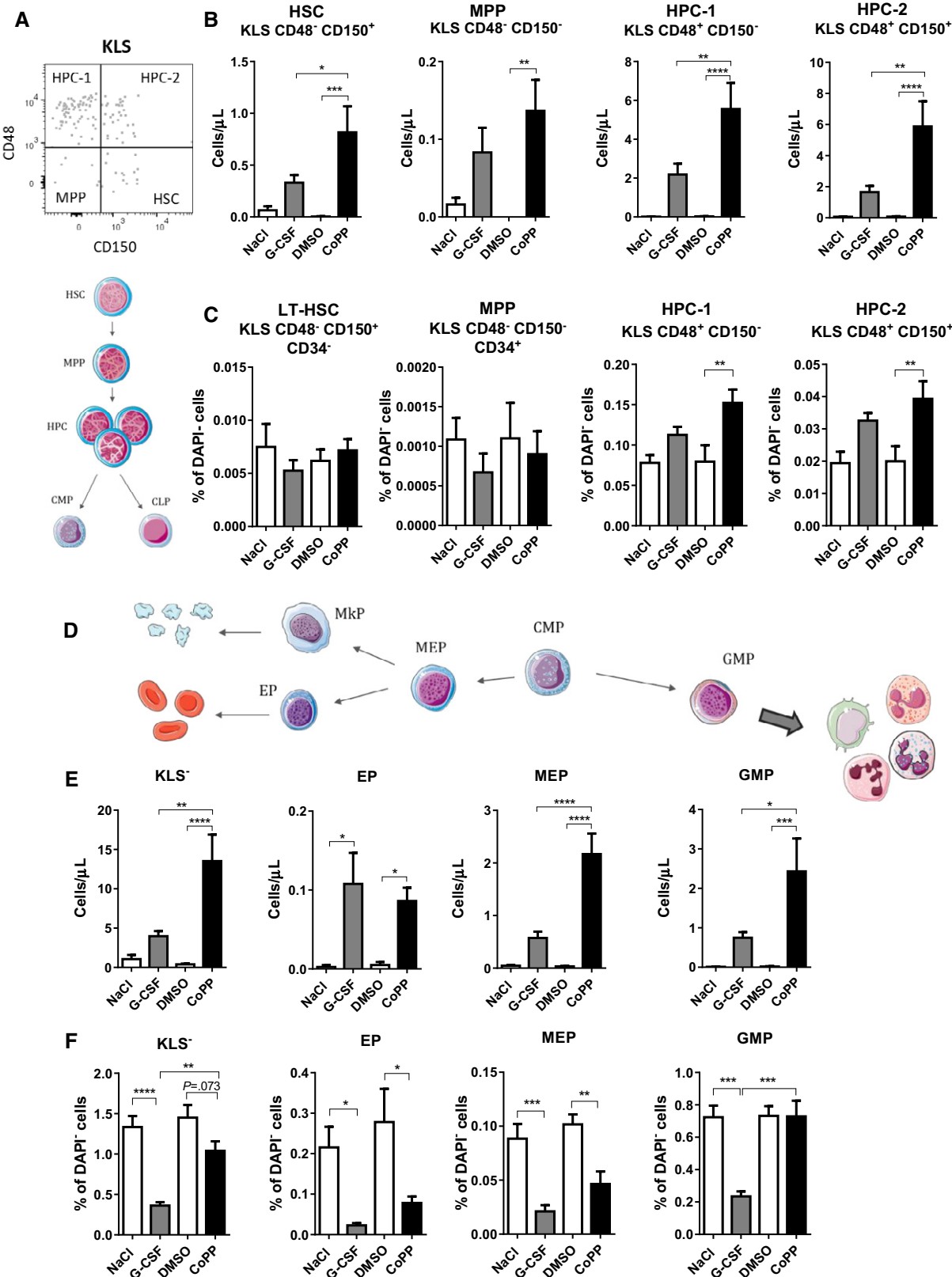

Figure 4.

◄

**Figure 4. CoPP mobilizes more HSPC than G-CSF in C57BL/6xFVB mice.**

A   Gating strategy of KLS (c-Kit$^+$Lin$^-$Sca-1$^+$) cells in blood, using CD48 and CD150 to distinguish HSC (hematopoietic stem cells), MPP (multipotent progenitors), and HPC (hematopoietic progenitors).

B   CoPP mobilizes higher numbers of HSC (KLS CD48$^-$CD150$^+$), MPP (KLS CD48$^-$CD150$^-$), HPC-1 (KLS CD48$^+$CD150$^-$), and HPC-2 (KLS CD48$^+$CD150$^+$) cells than G-CSF to the PB.

C   Treatment with CoPP increases percentage of KLS cells in BM, what is related to increase in HPC populations, but not in LT-HSC fraction. In contrast, frequency of MPP tends to decrease.

D   Scheme of common myeloid progenitor (CMP) differentiation toward granulocytes/monocytes, erythrocytes, and platelets.

E   The number of KLS$^-$ (c-Kit$^+$Lin$^-$Sca-1$^-$) is higher in mice treated with CoPP than in mice treated with G-CSF. Numbers of lineage-committed progenitors, GMP (granulocyte-macrophage progenitors, KLS$^-$ CD48$^+$CD150$^-$CD34$^+$) and MEP (megakaryocyte–erythroid progenitors, KLS$^-$ CD48$^+$CD150$^+$CD34$^-$), were higher after CoPP administration than after G-CSF. G-CSF and CoPP similarly increase EP numbers (erythrocyte progenitors, KLS$^-$ CD48$^-$CD150$^-$CD34$^-$).

F   Both G-CSF and CoPP treatments decrease the KLS$^-$ percentage in the BM; however, the decrease after CoPP is smaller. G-CSF and CoPP decrease percentages of MEP and EP, but only G-CSF decreases the percentage of GMP.

Data information: Results are shown as mean + SEM, one-way ANOVA with Bonferroni post-test, *n* = 7 mice per group. *$P \leq 0.05$; **$P \leq 0.01$; ***$P \leq 0.001$; ****$P \leq 0.0001$.

Donor mice (C57BL/6-Tg(UBC-GFP)30Scha/J) are on C57BL6 genetic background, which is described as poor HSPC mobilizing strain (Roberts *et al*, 1997). Thus, we analyzed in detail the mobilization in CoPP- and G-CSF-treated donors. As shown in Fig 5Ai,ii, CoPP was an effective mobilizer of KLS cells and granulocytes. In contrast, there was a bimodal distribution among G-CSF-treated mice, where only four out of 16 individuals efficiently mobilized KLS cells and granulocytes (Fig 5Ai,ii).

We analyzed hematological parameters and chimerism in recipients 2 weeks after transplantation. Although at that time point all recipient mice had WBC counts and platelet numbers below the physiological range (Fig 5B, gray box), we observed significantly higher mean numbers of WBC and platelets in mice that received CoPP-mobilized PBMC compared with non-treated mice (Fig 5B). Mice that received G-CSF had higher numbers of WBC and platelets than controls, but the difference was smaller and did not reach statistical significance. Mean hematocrit (HCT) levels were within the normal range already 2 weeks after PBMC transplantation in all groups (Fig 5B); however, only mice transplanted with CoPP-mobilized PBMC had significantly higher levels than the control mice. After 4 weeks, the WBC, platelet count, and HCT values reached the physiological range, without differences between the groups (Fig 5B). There were no differences between the G-CSF and CoPP groups in the numbers of lymphocytes, granulocytes, and monocytes, as well as RBC (Appendix Fig S4).

At all time points, we observed higher blood chimerism among total CD45$^+$ cells, granulocytes, and B cells in mice transplanted with CoPP-mobilized PBMC than in mice transplanted with G-CSF-mobilized PBMC. Chimerism among T cells tended to be higher in the G-CSF group 2 weeks after transplantation, but after 18 weeks tended to be higher in the CoPP group (Fig 5C).

To examine whether the hematological reconstitution was stable, 18 or 20 weeks after primary transplantation we performed secondary transplantation, using $10^7$ of whole BM cells to lethally irradiated secondary recipients. The mean GFP chimerism in KLS cells isolated from the BM of primary recipients was similar in the G-CSF and CoPP groups (Fig 5D). Fourteen weeks after the secondary transplantation, seven out of eight secondary recipients of CoPP-mobilized cells and three out of six secondary recipients of G-CSF-mobilized cells had at least 1% of GFP chimerism in the blood CD45$^+$ cells (Fig 5E, *P* = 0.24, Fisher's exact test, two-tailed). GFP chimerism was observed among all examined cell types

(Fig 5F). We observed GFP chimerism in BM KLS and KLS$^-$ cells in two out of six, and three out of eight, secondary recipients form G-CSF and CoPP groups, respectively (Fig 5G–J).

To sum up, CoPP induces mobilization of functional HSPC that upon transplantation rescue hematopoiesis in lethally irradiated mice, provides faster reconstitution during the first period after transplant, and produces higher blood chimerism than G-CSF-mobilized cells.

### Effect of CoPP on leukocytes depends on G-CSF, but not on HO-1/Nrf2 axis

G-CSF, IL-6, and MCP-1 were the most induced cytokines after CoPP administration (Figs 1 and 2). Given that role of G-CSF in mobilization is well established, we hypothesized that it is the main mediator of CoPP-induced mobilization and other cytokines, especially IL-6, are augmenting the effect. Therefore, we blocked G-CSF alone or in combination with IL-6 neutralizing antibodies before CoPP treatment and checked whether mobilization is diminished.

Efficient neutralization of G-CSF (Fig 6A) completely blocked the CD45$^+$ cell mobilization induced by CoPP and significantly reduced mobilization of granulocytes and HSC (Fig 6B). We did not observe any additional reduction of mobilization in mice treated with combination of G-CSF and IL-6 blocking antibodies.

To further study the role of IL-6 in CoPP-induced mobilization, we checked whether administration of G-CSF together with recombinant murine IL-6 (rmIL-6) will mimic the superior mobilization effect of CoPP. We did not observe any effect of simultaneous administration of G-CSF and rmIL-6 compared with G-CSF alone (Fig EV3A). However, 6 h after injection, IL-6 concentration in plasma was undetectable, whereas it was still present in the mice treated with CoPP (Fig EV3B).

As the induction of G-CSF expression by CoPP was not reported before, we performed qPCR and Luminex assay to investigate which tissues or organs respond to CoPP by producing G-CSF. We detected increased levels of mRNA for G-CSF in the lysates from the spleen, liver, and gastrocnemius muscle (Appendix Fig S5A). Luminex assay confirmed the detectable upregulation of G-CSF protein in the spleen (Appendix Fig S5B).

Next, we investigated whether the mechanism of CoPP action is dependent on Nrf2/HO-1 axis. CoPP is a known inducer of the *Hmox1* gene expression, and we expected that mobilizing properties

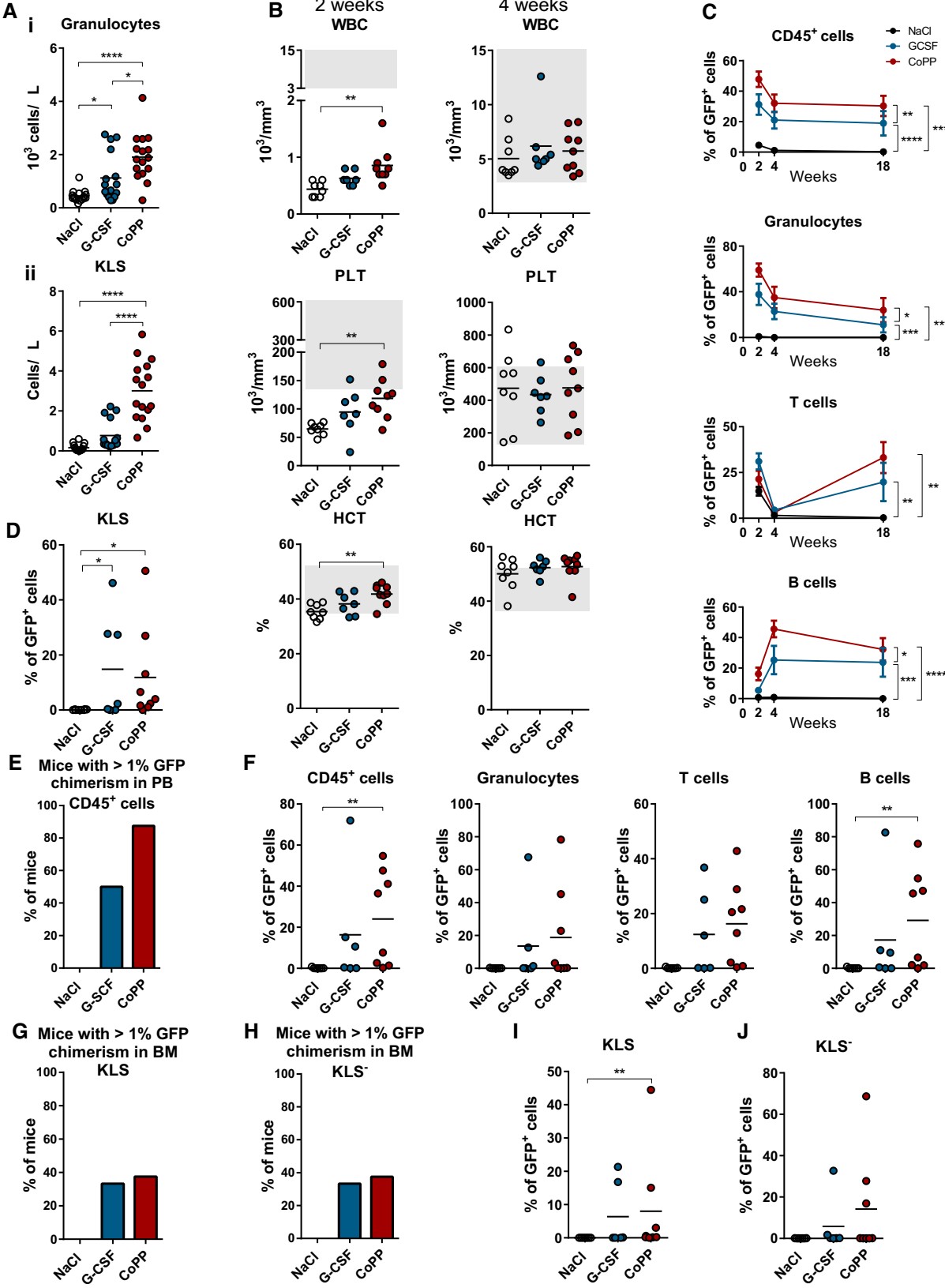

**Figure 5.**

Figure 5. CoPP mobilizes functional HSC that provide faster hematopoietic recovery and higher chimerism upon transplantation than G-CSF-mobilized HSC.

Donor GFP$^+$ mice were treated with NaCl, G-CSF, or CoPP daily. On the 5$^{th}$ day of the treatment, we transplanted $5 \times 10^6$ of isolated PBMC to the lethally irradiated GFP$^-$ recipient mice, together with $10^5$ GFP$^-$ BM-derived competitor cells. After 18 or 20 weeks, we performed secondary transplantation of primary recipients' BM to lethally irradiated secondary recipients and followed the chimerism for additional 14 weeks.

A    Effect of G-CSF and CoPP-induced mobilization in C57BL/6-Tg(UBC-GFP)30Sch/J donor mice (mean and individual values are shown, one-way ANOVA with Bonferroni post-test, $n = 16$ mice per group; *$P \leq 0.05$, ****$P \leq 0.0001$): Donor mice treated with CoPP had the highest number of mobilized granulocytes (i) and KLS cells (ii), compared with control or G-CSF-treated donor mice

B    Recipient mice which were transplanted with CoPP-mobilized PBMC had the highest number of WBC, PLT, and the highest hematocrit values 2 weeks after PBMC transplantation, compared with G-CSF-mobilized and control PBMC recipients (mean and individual values are shown, one-way ANOVA with Bonferroni post-test, NaCl: $n = 8$, G-CSF: $n = 7$, CoPP: $n = 9$ mice per group; **$P \leq 0.01$).

C    Recipient mice which were transplanted with CoPP-mobilized PBMC had the highest chimerism among CD45$^+$ cells, granulocytes and B cells 2, 4, and 18 weeks after transplantation. Chimerism among T cells was the highest in recipients of G-CSF-mobilized PBMC 2 weeks after transplantation, but 18 weeks after transplantation it was the highest in CoPP-mobilized PBMC recipients (mean + SEM, two-way ANOVA with Bonferroni post-test, NaCl: $n = 8$, G-CSF: $n = 7$, CoPP: $n = 9$ mice per group; *$P \leq 0.05$; **$P \leq 0.01$; ***$P \leq 0.001$; ****$P \leq 0.0001$).

D    The majority of primary recipients transplanted with G-CSF and CoPP-mobilized PBMC have detectable GFP chimerism among BM KLS cells (mean and individual values are shown, Kruskal–Wallis test with Dunn's post-test, NaCl: $n = 8$, G-CSF: $n = 7$, CoPP: $n = 9$ mice per group; *$P \leq 0.05$).

E    Fraction of secondary recipient mice with chimerism in PB CD45$^+$ cells exceeding 1% tends to be higher after CoPP-mobilized PBMC transplant than G-CSF-mobilized PBMC transplant.

F    PB chimerism in secondary recipients is higher in the CoPP group than in the control group (mean and individual values are shown, Kruskal–Wallis test with Dunn's post-test, NaCl: $n = 7$, G-CSF: $n = 6$, CoPP: $n = 8$ mice per group; **$P \leq 0.01$).

G, H    Percentage of secondary recipient mice with GFP chimerism in BM KLS (G) and KLS$^-$ (H) cells higher that 1% is similar in the G-CSF and CoPP groups.

I, J    GFP chimerism in KLS (I) and KLS$^-$ (J) cells in the BM of secondary recipients (mean and individual values are shown, Kruskal–Wallis test with Dunn's post-test, NaCl: $n = 7$, G-CSF: $n = 6$, CoPP: $n = 8$ mice per group; **$P \leq 0.01$).

of CoPP are mediated by HO-1 activation. To verify this supposition, we checked if CoPP induces mobilization in HO-1$^{-/-}$ mice.

As HO-1$^{-/-}$ individuals have increased susceptibility to the toxic effect of many compounds, in the experiment with HO-1$^{+/+}$ and HO-1$^{-/-}$ mice we administered CoPP three times, every second day (instead of daily for 5 days). We observed mobilization of total WBC and granulocytes in both CoPP-treated groups, despite the higher basal number of granulocytes in HO-1$^{-/-}$ individuals (Fig 6C,D). Accordingly, CoPP led to elevated plasma G-CSF, IL-6, and MCP-1 both in HO-1$^{+/+}$ and HO-1$^{-/-}$ mice (Fig 6E). Thus, CoPP-induced mobilization is not dependent on HO-1 induction.

Having excluded involvement of HO-1 in CoPP-induced mobilization, we examined the upstream signaling. The expression of *Hmox1* gene in response to CoPP is regulated by the Nrf2 transcription factor. To verify whether Nrf2 is involved in the CoPP-induced mobilization, we treated Nrf2$^{+/+}$ and Nrf2$^{-/-}$ mice with CoPP daily for 5 days.

Cobalt protoporphyrin increased the number of CD45$^+$ cells and percentage of granulocytes in the blood of either Nrf2$^{+/+}$ or Nrf2$^{-/-}$ mice (Fig 6F,G). In both groups, we observed similarly elevated plasma G-CSF and IL-6 after CoPP treatment, while the increase in MCP-1 concentration was even higher in Nrf2$^{-/-}$ mice (Fig 6H). Percentage of monocytes was not affected by CoPP; however, regardless of the treatment, it was higher in Nrf2$^{-/-}$ mice than in Nrf2$^{+/+}$ mice (Appendix Fig S6).

These results indicate that mechanism of CoPP-induced mobilization is dependent on G-CSF, but not on Nrf2/HO-1 axis.

## Discussion

Recombinant G-CSF is commonly used to mobilize and harvest HSCs for transplantation (To *et al*, 2011). It is also applied in the treatment of various types of neutropenia (Gabrilove *et al*, 1988; Hoggatt & Pelus, 2014). Although G-CSF was successfully introduced to clinical practice, there are cases when G-CSF treatment remains inefficient

(Wuchter *et al*, 2010; To *et al*, 2011). One of the proposed strategies to overcome these limitations would imply induction of endogenous G-CSF expression by drug administration rather than administration of recombinant G-CSF itself (Hoggatt & Pelus, 2014).

In this study, we demonstrate that CoPP induces G-CSF and mobilizes HSC and myeloid cells into the peripheral circulation. We showed that CoPP-induced mobilization has advantages over G-CSF administration when used to obtain mobilized PB for subsequent transplantation to irradiated host. First, CoPP administration mobilizes a higher number of HSC. Second, transplantation of PBMC from CoPP-treated donors leads to faster hematopoietic reconstitution in first 2 weeks after transplantation—a critical period when the immunocompromised recipient needs the fastest possible hematopoietic recovery. This is likely due to the higher number of progenitors mobilized by CoPP. In contrast to G-CSF, CoPP does not increase the number of T cells. This could be clinically relevant, as T cells may cause GVHD (Zhang *et al*, 2005; Vasu *et al*, 2016; MacDonald *et al*, 2017) and impair engraftment of HSC upon transplantation (Müller *et al*, 2010). Finally, transplantation of PBMC from CoPP-treated donors resulted in a higher long-term blood chimerism, even upon secondary transplantation. While the first blood cells are produced by short-lived progenitors, the level of the long-term chimerism in transplantation assays is linked to long-term HSC (Purton & Scadden, 2007).

Moreover, CoPP mobilizes a partially different pool of cells than G-CSF. Although CoPP mobilizes granulocytes with similar efficiency to G-CSF, their phenotypes are distinct. G-CSF-mobilized granulocytes are characterized by lower granularity and lower Ly6G expression, resembling immature cells, while CoPP-mobilized granulocytes resemble more mature cells with higher granularity and higher Ly6G expression. This feature might be of great importance in treating neutropenia, especially chemotherapy-induced neutropenia. It was shown that dexamethasone/G-CSF-mobilized granulocytes of less mature phenotype, although able to produce normal ROS levels after stimulation, have less developed granules and are less efficient in the killing of *Candida albicans* (Gazendam *et al*, 2016). If confirmed

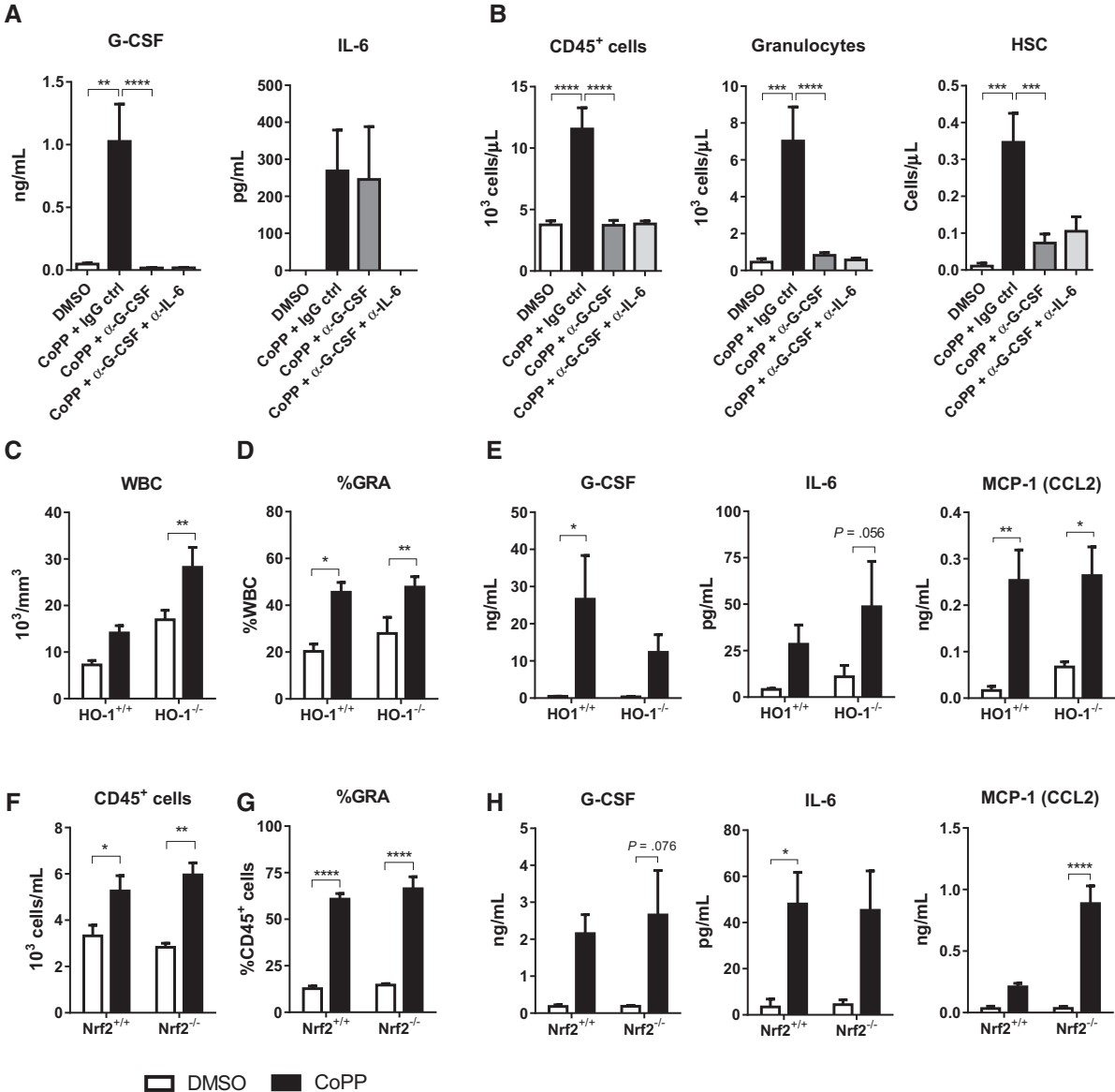

**Figure 6. CoPP-induced mobilization depends on G-CSF, but not depends on Nrf2/HO-1 pathway.**

A, B  Wild-type C57BL/6 mice were injected with anti-G-CSF antibody alone or in combination with anti-IL-6 antibody 1 h prior to CoPP treatment; the injections were repeated daily for 5 days, and mice were sacrificed 6 h after the last CoPP injection. (A) G-CSF and IL-6 concentration in plasma. (B) CD45+ cells, granulocytes, and HSC numbers in PB measured by flow cytometry (DMSO: $n = 4$, CoPP + IgG ctrl: $n = 6$, CoPP + α-G-CSF: $n = 9$, CoPP + α-G-CSF + α-IL-6: $n = 7$).

C–E  HO-1-deficient and control C57BL/6xFVB mice were injected with CoPP three times, every second day and sacrificed 24 h after last injection. (C) Total leukocyte counts in PB (HO-1+/+: $n = 5$, HO-1−/− DMSO: $n = 6$, HO-1−/− CoPP: $n = 4$ mice per group). (D) Granulocyte percentage among PB leukocytes (HO-1+/+: $n = 5$, HO-1−/− DMSO: $n = 6$, HO-1−/− CoPP: $n = 4$ mice per group). (E) Cytokine and growth factor concentrations in plasma (HO-1+/+ $n = 5$, HO-1−/− DMSO: $n = 5$, HO-1−/− CoPP: $n = 3$ mice per group).

F–H  Nrf2-deficient and control C57BL/6 mice were injected with CoPP daily for 5 days and sacrificed 6 h after last injection (DMSO: $n = 4$, CoPP: $n = 5$ mice per group). (F) CD45+ cell numbers in PB measured by flow cytometry. (G) Granulocyte percentage among PB leukocytes. (H) Cytokine and growth factor concentrations in plasma.

Data information: Results are shown as mean + SEM, one-way ANOVA with Bonferroni post-test (A, B) or two-way ANOVA with Bonferroni post-test (C–H). *$P \leq 0.05$; **$P \leq 0.01$; ***$P \leq 0.001$; ****$P \leq 0.0001$.

in further studies, that feature of CoPP could be potentially desirable, as *C. albicans* infections are one of the most frequent complications after chemotherapy (Teoh & Pavelka, 2016).

While using CoPP as a mobilizing drug was not reported before, two published studies (Konrad *et al*, 2014; Wysoczynski *et al*, 2015)

demonstrate how CoPP or SnPP modulates the effect of other mobilizing factors. Konrad *et al* (2014) showed that pretreatment of mice with CoPP decreased mobilization of neutrophils and their migration to the lungs induced by LPS. In the report by Wysoczynski *et al* (2015), SnPP injected together with G-CSF or AMD3100 potentiated

their mobilizing effect on HSPC, but given alone did not affect the mobilization of WBC (Wysoczynski *et al*, 2015). In our study, SnPP does not have any effect on measured parameters. Of course, the effect of CoPP or SnPP injected alone can differ when porphyrins are administered together with other factors. However, although results of this study do not contradict the above findings (Konrad *et al*, 2014; Wysoczynski *et al*, 2015), our previous research using HO-1$^{-/-}$ mice suggested that HO-1 deficiency may impair G-CSF-induced mobilization (Bukowska-Strakova *et al*, 2017).

The question arises as to what mechanism is responsible for the observed difference between CoPP- and G-CSF-induced mobilization. G-CSF is crucial for neutrophil egress from the BM in steady-state conditions, but the expression of G-CSFR on neutrophils (Semerad *et al*, 2002) and hematopoietic progenitor cells (Liu *et al*, 2000) is not necessary for their mobilization. While we show that the major CoPP-induced factor responsible for mobilization is G-CSF, CoPP also induces other cytokines including IL-6, MCP-1, and IP-10. These mediators are not induced by administration of G-CSF, implying that they are not downstream of G-CSF signaling. It is possible that they are co-responsible with G-CSF for CoPP-induced mobilization. Indeed, IL-6 was shown to be crucial cytokine produced by HSPC to govern myeloid differentiation (Zhao *et al*, 2014) and is able to induce mobilization of granulocytes (Pojda & Tsuboi, 1990). We could not mimic the effect of CoPP by simultaneous injection of G-CSF with IL-6. However, the induction of endogenous IL-6 by CoPP seems to be more efficient than injecting recombinant IL-6—6 h after the last CoPP treatment IL-6 was still detectable in plasma, whereas it was undetectable in the mice treated with rmIL-6.

The most known pharmacological property of CoPP is the activation of the Nrf-2/HO-1 axis (Shan *et al*, 2006). As we and others have shown previously, HO-1 regulates myeloid cell development and these observations initially triggered us to study the possible application of CoPP to modulate myelogenesis. Although we did observe activation of HO-1 upon CoPP administration, this increase was not a causative factor here. Using HO-1- and Nrf-2-deficient mice, we demonstrated that CoPP-induced mobilization is largely independent of the Nrf-2/HO-1 axis. This observation also indicates that results obtained after CoPP administration and ascribed to the effects of HO-1 induction should be interpreted carefully.

We think that it would be advantageous not to test the new compounds with a potential clinical application on one strain with a uniform genetic background. Knowing that mobilization efficiency in response to G-CSF differs between strains and that crosses between strains increase variability in response to G-CSF (Roberts *et al*, 1997; Hasegawa *et al*, 2000), after our initial and unexpected observation of CoPP-induced mobilization in the C3H strain, we decided to test the CoPP mobilization in different strains. Our reasoning was that CoPP would have an advantage over G-CSF if it were able to efficiently mobilize different strains of mice, especially the "poor mobilizers", such as C57BL/6. Using such an approach, we eliminated the risk that CoPP acts only in a strain-specific manner.

We think that CoPP or its derivative could be used as a drug for mobilization purposes. Other metalloporphyrins were already tested in clinical trials as inhibitors of HO-1 activity in neonatal bilirubinemia (Kappas *et al*, 1988, 2001; Martinez *et al*, 1999), proving the safety use of this group of compounds. Various porphyrins are also widely used in the photodynamic therapy in cancer (O'Connor *et al*, 2009). The phototoxicity of protoporphyrins that is desirable in photodynamic therapy would be considered a side effect when used as mobilization factor. However, the phototoxic properties differ between protoporphyrins (Scott *et al*, 1990). Our results indicate that CoPP has negligible ROS production upon laser photolysis, in contrast to SnPP which shows a very high quantum efficiency of singlet oxygen generation (Appendix Fig S7). Therefore, we do not expect that phototoxicity would exclude the potential clinical application of CoPP. In our preliminary studies, we did not observe weight loss (data not shown) or the upregulation of general pro-inflammatory cytokines like TNFα and IL-1 (Appendix Fig S8A) in mice treated with CoPP at the dose of 10 mg/kg, indicating there is no acute toxicity, although some changes in liver and muscle enzymes were visible (Appendix Fig S8B). The cells that were mobilized by CoPP proved their viability and functionality in the transplantation model. Potential future clinical application of CoPP would require testing a range of CoPP doses, its pharmacokinetics, and bioavailability. While many protoporphyrins are not orally absorbed, application of lipid-based formulations allows for oral absorption (Fujioka *et al*, 2016), what would provide further benefit of using CoPP in therapy.

To conclude, CoPP represents a new mobilizing factor that may potentially translate into clinical applications and overcome the limitations of current mobilization strategies.

# Materials and Methods

### Mice

Animal work was done in accordance with the good animal practice and approved by the First or Second Local Ethical Committee for Animal Research at the Jagiellonian University (approval numbers: 106/2007, 180/2014, 8/2015, 28/2015, 276/2018, and 90/2019).

C57BL/6xFVB HO-1$^{-/-}$ mice were initially provided by Dr. Anupam Agarwal, University of Alabama, Birmingham, USA. HO-1$^{-/-}$ mice were initially created on the 129/Sv × C57BL/6 background (Poss & Tonegawa, 1997) and backcrossed to C57BL/6 mice. Due to the very poor breeding, they were crossed to a well-breeding FVB strain (Kapturczak *et al*, 2004). Resulting HO-1$^{+/-}$ offspring were backcrossed to FVB WT several times, which resulted in mixed C57BL/6 × FVB background. Crossing the HO-1$^{-/-}$ to FVB background led to the improved breeding (1 HO$^{-/-}$ pup out of 20), while retaining the original HO-1$^{-/-}$ characteristics (Kapturczak *et al*, 2004).

Nrf2-deficient C57BL/6 mice (Nrf2$^{-/-}$) generated by Itoh *et al* (1999) were kindly provided by Prof. Antonio Cuadrado (Universidad Autonoma de Madrid, Spain). C3H mice were purchased from Charles River. C57BL/6-Tg(UBC-GFP)30Scha/J were bought from the Jackson Laboratories.

Animals in each cage were randomly assigned to all experimental groups.

Cobalt protoporphyrin mobilization experiment in HO-1$^{-/-}$ mice was performed in the conventional animal facility. All other experiments were performed in specific pathogen-free (SPF) conditions, with constant light/dark cycle (14/10 h) and continuous monitoring of temperature and humidity. Mice were kept in groups ≤ 5 in the individually ventilated cages with food and water *ad libitum*.

## Mobilization experiments

Cobalt protoporphyrin and SnPP (Frontier Scientific) were dissolved in DMSO (Sigma-Aldrich) at concentration of 0.2 g/ml. The stock solutions were diluted 160× in 0.9% NaCl solution. Mice were injected intraperitoneally (i.p.) at the dose of 10 mg/kg (15 μmol/kg). Recombinant human G-CSF (rhG-CSF; Amgen) was used at the dose of 250 μg/kg. Murine recombinant IL-6 (R&D) was used in concentration 50 μg/kg. Compounds were injected i.p. daily for 5 days or three times, every second day, depending on the experiment, as indicated in the text and Fig EV1. PB, spleen, and BM were collected 6 or 24 h after the last injection of mobilizing factors. Complete blood count was done using Vet abc Plus+ analyzer (Horiba).

## *In vivo* blocking of cytokines with neutralizing antibodies

To block G-CSF and IL-6 in mice, we used monoclonal rat IgG$_1$ antibodies: anti-G-CSF (R&D, cat #MAB414, clone 67604) and anti-IL-6 (BioXcell, clone MP5-20F3). Rat IgG$_1$ anti-HRP (BioXcell, clone HRPN) was used as a IgG isotype control. The antibodies were administered 1 h before each CoPP dose, for five consecutive days. The experimental groups included the following: DMSO (no CoPP, no antibodies), CoPP + IgG control, CoPP + anti-G-CSF, and CoPP + anti-G-CSF + anti-IL-6. Each single dose was anti-G-CSF 12.5 μg/mice, anti-IL-6 130 μg/mice, and IgG control 142.5 μg/mice. Six hours after the last treatment, samples were collected for analysis (Fig EV1).

## Heme oxygenase activity measurement

Liver tissue fragments were frozen at −80°C immediately after dissection. HO activity in tissue homogenates (CO produced within 15 min after adding the reaction substrates NADPH and methemalbumin) was measured using gas chromatography (Vreman & Stevenson, 1988).

## Cytokine concentration analysis

Luminex assays (MILLIPLEX MAP Mouse Cytokine/Chemokine Premixed 32-Plex, Mouse Cytokine/Chemokine Magnetic Bead Premixed 32-Plex and the custom assay panel, Millipore; Custom Premixed Luminex Mouse Magnetic Assay, 6-Plex, R&D) were performed on −80°C frozen plasma samples according to manufacturer's instructions, and the signal was detected using FLEXMAP 3D system (Millipore).

## Mobilized blood transplantation

Two- to three-month-old female C57BL/6 recipient mice were irradiated with $^{137}$Cs γ source at 110 cGy/min (2 × 450 cGy at 4-h intervals) 24 h before the transplantation. Two- to three-month-old female C57BL/6-Tg(UBC-GFP)30Scha/J donor mice were treated for five consecutive days with CoPP, rhG-CSF, or NaCl as described above. Six hours after the last injection, blood was collected and PBMC were isolated using Ficoll-Paque PLUS (GE Healthcare). $5 \times 10^6$ of GFP$^+$ PBMC with $1 \times 10^5$ GFP$^-$ BM-derived competitor cells were injected into the tail vein of irradiated recipients.

### The paper explained

**Problem**

Pharmacological mobilization is mainly used in two situations: (i) treatment of chemotherapy-induced neutropenia; and (ii) mobilization of HSCs for their transplantation. Chemotherapy is often the only therapeutic option for cancer patients. Unfortunately, chemotherapy comes with significant side effects, toxicity to the hematopoietic system being one of the most dangerous. HSC transplantation is the live-saving therapy for various hematological disorders. For many years now, mobilized blood has been the preferred source of HSCs for transplantation instead of BM biopsy. Recombinant G-CSF (filgrastim) is the most widely used mobilizing factor. However, up to 30% healthy donors and 60% high-risk patients fail to respond to mobilization (To *et al*, 2011). In many cases, existing alternatives (pegfilgrastim, plerixafor) help to overcome the G-CSF ineffectiveness but, due to their high cost, are sometimes unaffordable for the patients.

**Results**

We have found that cobalt protoporphyrin (CoPP) induces endogenous G-CSF and other cytokines and may have superior mobilizing properties in comparison with the currently used drug, recombinant G-CSF. First, CoPP-mobilized PB provides faster reconstitution and higher long-term blood chimerism upon transplantation than G-CSF-mobilized blood. Second, CoPP mobilizes a higher number of HSCs than G-CSF, but in contrast to G-CSF, CoPP does not mobilize T cells, which could be important in graft-versus-host disease prevention. Finally, CoPP mobilizes granulocytes with a more mature phenotype, which could find some application for the treatment of neutropenia.

**Impact**

Our data suggest that CoPP could be used for the treatment of hematological disorders. CoPP can work more efficiently than G-CSF. Moreover, as a simple compound it should be much cheaper in production, making the treatment available for a broader group of patients. The possibility of developing an orally absorbed CoPP formulation may further facilitate its potential clinical use.

Fourteen weeks after the primary transplantation, $10 \times 10^6$ of BM cells were transplanted to the secondary recipients (Fig EV2). PB chimerism was analyzed in B cells, T cells, and granulocytes as shown in Appendix Fig S9.

## Flow cytometry

The cell suspensions were filtered with 70 μm strainer, depleted of erythrocytes by a hypotonic solution, washed, and stained in PBS 2% FBS for 20 min on ice. Samples were collected using LSR II and LSR Fortessa cytometers (BD) and analyzed using BD FACS-Diva and FlowJo software. The absolute numbers of cells/μl were calculated using BD Trucount Tubes (BD) and BD FACS lysing solution (BD). Antibodies for flow cytometry were used at 1:50 or 1:100 dilutions. Flow cytometry reagents and gating strategies are shown in Appendix Table S1 and Appendix Figs S9–S14.

## ROS production assay

Mice were treated with CoPP, G-CSF, DMSO, and NaCl for 5 consecutive days as described above. Six hours after the last injection, blood samples were collected. Phagoburst assay (Glycotope Biotechnology) was performed according to manufacturer's instructions.

## Heatmaps

Heatmaps were created using heatplot function (Eisen *et al*, 1998), gplots (Warnes *et al*, 2016), and made4 package from Bioconductor (Culhane *et al*, 2005, 4) in R version 3.2.5 (R Core Team, 2015) using R studio version 0.99.451 (RStudio Team, 2015). For Fig 1C, base-10 logarithms of raw data were used, and for Fig 2B, raw data were used for creating heatmaps.

## viSNE

$CD11b^+CD11c^-$ cells from five samples/group were concatenated and subsampled to 5000 events. viSNE plots were created using Cyt (Amir *et al*, 2013) in MATLAB.

## qPCR

Quantitative real-time PCR (qPCR) was performed using SYBR® Green JumpStart™ Taq ReadyMix™ (Sigma-Aldrich) on StepOne Plus thermocycler (Applied Biosystems) and analyzed using StepOne software. Commercially available primers *M-Csf3_2* Sigma predesigned KiCqStart™ Primers spanning exons 4–5 were used, with annealing temperature of 60°C.

## Quantum yield of singlet oxygen formation

To determine the quantum yields of singlet oxygen formation ($\Phi_\Delta$), the laser flash photolysis (LKS 60 Applied Photophysics) was applied. The $\Phi_\Delta$ values were determined using the comparative method, which relies on the detection of $^1O_2$ phosphorescence at $\lambda = 1{,}270$ nm with phenalenone (Sigma) as a standard. Investigated materials were dissolved in ethanol at the concentration necessary to produce an absorbance of 0.25 at the excitation wavelength ($\lambda_{exc} = 355$ nm). To detect the decay of singlet oxygen emission, the excitation of sample was generated using the third harmonic ($\lambda_{exc} = 355$ nm) of a Nd/YAG laser (20 Hz Brilliant, Quantel). We analyzed the intensity of singlet oxygen emission at $t = 0$ as a function of the relative energy of laser pulses. The $\Phi_\Delta$ value was calculated using the following formula:

$$\Phi_\Delta = \Phi_\Delta^{ref} \cdot (a/a_{ref})$$

where $\Phi_\Delta^{ref}$ is a quantum yield of a reference sample, i.e., $\Phi_\Delta^{ref} = 0.95$ in ethanol (Schmidt *et al*, 1994), while $a$ and $a_{ref}$ denote the slope determined from the intensity of $^1O_2$ phosphorescence plotted as a function of laser energy for investigated material and phenalenone, respectively.

## Statistical analysis

In our experiments, we were not able to prespecify the expected effect size; therefore, no statistical power and sample size estimation were done. We used the maximal number of animals per experimental group that were allowed by local ethical committee. Nevertheless, we performed independent experiments to confirm majority of our conclusions. Animals in each cage were randomly assigned to all experimental groups. Injections were not done in a blinded manner, as CoPP is colored. All samples were collected and analyzed in blinded manner. Statistical analysis was done with GraphPad Prism software that estimates variance equality for analyzed datasets. Data are presented as mean with SEM or mean with individual values. For multigroup comparisons with one variable, we used one-way ANOVA with additional post-tests, and for multigroup comparisons with two variables, we used two-way ANOVA with additional post-tests. Grubbs' test was used to determine significant outliers. Results were considered as statistically significant, when $P \leq 0.05$ (*$P \leq 0.05$; **$P \leq 0.01$; ***$P \leq 0.001$; ****$P \leq 0.0001$). Exact $P$ values for all the comparisons together with the statistical tests used are listed in Appendix Table S2.

## Schemes

Schemes were created using images from Servier Medical Art https://smart.servier.com/ with modifications.

**Expanded View** for this article is available online.

## Acknowledgements

The authors would like to acknowledge Ewa Werner, Karolina Hajduk, Jan Morys, Sylwester Mosiołek, and Izabella Skulimowska for their help and support in performing animal experiments; Agnieszka Andrychowicz-Róg for technical support; Dr. Jacek Kijowski for help with BM transplantations in mice; and Dr. Marcin Surmiak for help with Luminex assay. The study was supported by structural funds from the EU (grant 01.02.00-069/09) and by the National Science Center (Grant Harmonia no NCN2015/18/M/NZ3/00387 awarded to A.J. and Grant Preludium no NCN2013/11/N/NZ3/00956 awarded to K.S.). A.S. and K.S. were supported by Mobility Plus grants financed by the Ministry of Science and Higher Education, Republic of Poland (grants no 1319/MOB/IV/2015/0 and 1273/MOB/IV/2015/0). The Faculty of Biochemistry, Biophysics, and Biotechnology of Jagiellonian University is a partner of the Leading National Research Center (KNOW) supported by the Ministry of Science and Higher Education.

## Author contributions

AS, KS, and AJ designed the experiments, AS, KS, WNN, KB-S, LM, MG, MŻ, MC, NK-T, and MR-B performed the experiments, AS and KS analyzed the data, WNN, KB-S, AR, and JD assisted with experimental designs and interpretation, AS, KS, and AJ wrote the manuscript.

## Conflict of interest

AS, KS, AJ, and JD are the coinventors of the patents US10010557 and US10328085: *Cobalt porphyrins for the treatment of blood-related disorders* and EP 3139917: *Cobalt protoporphyrin IX for the treatment of blood-related disorders* granted to the Jagiellonian University. The remaining authors declare no conflict of interest.

## For more information

Lab website: http://nicheworks.eu/
Department website: https://zbm.wbbib.uj.edu.pl/en_GB/

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
