## [Review Process File · EMBO Molecular Medicine]

Cobalt protoporphyrin IX increases endogenous G-CSF and mobilizes HSC and granulocytes to the blood

Agata Szade, Krzysztof Szade, Witold N. Nowak, Karolina Bukowska-Strakova, Lucie Muchova, Monika Gońska, Monika Żukowska, Maciej Cieśla, Neli Kachamakova-Trojanowska, Marzena Rams-Baron, Alicja Ratuszna, Józef Dulak, Alicja Józkowicz

Review timeline:

Submission date:	18 July 2018
Editorial Decision:	6 September 2018
Revision received:	6 September 2019
Editorial Decision:	25 September 2019
Revision received:	9 October 2019
Accepted:	15 October 2019

Editor: Céline Carret

Transaction Report:

1st Editorial Decision

6 September 2018

Thank you for the submission of your manuscript to EMBO Molecular Medicine. We have now heard back from the two referees whom we asked to evaluate your manuscript.

You will see from the comments below that both referees find the paper of interest and clinically relevant. However, we would like to strongly encourage you to address the mechanism as recommended. Upon our cross-commenting exercise, both referees fully agreed with one another and the use of the commercially KO mice was advanced as the best way to address the mechanism. Please also make sure to answer all the other items.

We would therefore welcome the submission of a revised version within three months for further consideration and would like to encourage you to address all the criticisms raised as suggested to improve conclusiveness and clarity. Please note that EMBO Molecular Medicine strongly supports a single round of revision and that, as acceptance or rejection of the manuscript will depend on another round of review, your responses should be as complete as possible.

I look forward to receiving your revised manuscript.

***** Reviewer's comments *****

Referee #1 (Remarks for Author):

In this paper, Szade and colleagues present a series of *in vivo* mouse experiments showing that cobalt protoporphyrin (CoPP), previously shown to *a.o.* activate the Nrf-2/HO-1 axis, is a strong inducer of hematopoietic stem cell (HSC) and granulocyte mobilization *in vivo*. Nowadays, HSC mobilization for transplantation purposes in the clinic is being induced by treatment of donors with G-CSF or alternatively, in infrequent cases where G-CSF does not work, with the CXCR4 antagonist Plerixafor. In this paper, it is convincingly shown that CoPP is as effective as G-CSF in mobilizing HSCs, and that CoPP-mobilized HSCs are functionally fit and perform well in transplantation settings. The data presented are novel and of interest. The experiments have been carefully performed and the authors propose a plausible model as to how CoPP may be involved in HSC and granulocyte mobilization via a mechanism involving the release of G-CSF and to a lesser extent IL-6 and MCP-1 (Figure 6I).

Major comments:

1. A weaker aspect of the study is that a mechanistic explanation for this model is lacking. It remains unclear which cells or tissues respond to CoPP by increasing their G-CSF production/release and neither has it been shown in G-CSF or G-CSFR deficient models that the effects of CoPP indeed are mainly driven by G-CSF as suggested in Figure 6I. One obvious way to approach this is to study the effects of CoPP in G-CSF^{-/-} or G-CSFR^{-/-} mice. The authors are clearly aware of this option; in their discussion on p15/16 they provide a quite lengthy but not very convincing argumentation as to why these experiments were not done. Given that Csf3 and Csf3r knockout strains are commercially available, such experiments would be doable in a relatively short time frame.
2. Another aspect that remains unanswered is the link with the CXCL12/CXCR4 axis, the other well-known pathway involved in stem cell mobilization/homing which is altered therapeutically with Plerixafor to mobilize HSCs in G-CSF nonresponsive patients. Although CXCL12 (SDF1) does not appear to show up in the Luminex-based serum assays in CoPP-treated mice (Figure 1E), this does not exclude a possible role of the CXCL12/CXCR4 axis in CoPP-induced HSC mobilization. If this would be the case, CoPP might work as an alternative for Plerixafor/AMD3100. Again, this possibility could easily be explored in commercially available Cxcl12 or Cxcr4 deficient mice.

Minor comments:

1. Introduction on CoPP-induced HO-1 enzyme activity is largely superfluous and can be condensed significantly.
2. Same applies to the extensive argumentation as to why G-CSF and IL-6 knockout strains were not used in the study (related to major comments above)
3. A few typo's also in Figures, e.g., G-SCF in Figure 6D, need attention.

Referee #2 (Remarks for Author):

This is an interesting manuscript by Szade and colleagues assessing the ability of CoPP to mobilize cells from the bone marrow, including HSPCs and granulocytes. This work has clinical implications in regard to patients with neutropenia, which leads to numerous detrimental consequences, and for patients in need of HSPC transplantation. It is also interesting that this process is independent of the Nrf1/HO-1 axis. Please see below additional comments / questions to be addressed by the authors.

- 1) In the experiments, the authors assess cell mobilization at different time points, for example in Figure 1 - 24 hours, and Figure 2 - 6 hours. Is there a rationale for varying the time point of analysis? The authors are also assessing different genetic background mice. Does the use of different mice reflect the variation in timing of harvest for analysis?

2) The authors feel that the CoPP effect for mobilization of HSPCs and granulocytes seems to be related to G-CSF and additional cytokines, such as IL-6. The authors discuss the challenges of confirming these findings using G-CSF or IL-6 deficient mice. However, to provide further insight into this hypothesis, have the authors tried to inject mice with a combination of G-CSF and cytokines (such as IL-6), to see if these mice respond in a manner more analogous to CoPP?

3) The authors demonstrate CoPP induces mobilization of HSPCs that upon transplantation into irradiated mice have faster reconstitution and higher blood chimerism compared with cells mobilized by G-CSF. Do the authors know the mechanism for this improved response? Did the authors assess the cytokine panel (Figure 1) in irradiated mice that received HSPCs from CoPP and G-CSF mobilized cells? Can the authors give additional insight into this beneficial response?

1st Revision - authors' response

6 September 2019

We are very grateful to the Reviewers for their comments and suggested improvements to the manuscript. We are glad that our article was found to be of interest and potentially relevant to the clinical field. We fully agree with the main remark that our findings were lacking the mechanistic explanation and we concentrated our efforts to demonstrate the crucial molecular mediator of CoPP-induced mobilization.

We would also like to thank the Reviewers for allowing us to use the G-CSF blocking antibodies as the G-CSF KO mice turned out to be unavailable.

Below please find point-by-point response to Reviewers' comments.

Referee #1 (Remarks for Author): *In this paper, Szade and colleagues present a series of in vivo mouse experiments showing that cobalt protoporphyrin (CoPP), previously shown to a.o. activate the Nrf-2/HO-1 axis, is a strong inducer of hematopoietic stem cell (HSC) and granulocyte mobilization in vivo. Nowadays, HSC mobilization for transplantation purposes in the clinic is being induced by treatment of donors with G-CSF or alternatively, in infrequent cases where G-CSF does not work, with the CXCR4 antagonist Plerixafor. In this paper, it is convincingly shown that CoPP is as effective as G-CSF in mobilizing HSCs, and that CoPP-mobilized HSCs are functionally fit and perform well in transplantation settings. The data presented are novel and of interest. The experiments have been carefully performed and the authors propose a plausible model as to how CoPP may be involved in HSC and granulocyte mobilization via a mechanism involving the release of G-CSF and to a lesser extent IL-6 and MCP-1 (Figure 6I). Major comments: 1. A weaker aspect of the study is that a mechanistic explanation for this model is lacking. It remains unclear which cells or tissues respond to CoPP by increasing their G-CSF production/release and neither has it been shown in G-CSF or G-CSFR deficient models that the effects of CoPP indeed are mainly driven by G-CSF as suggested in Figure 6I. One obvious way to approach this is to study the effects of CoPP in G-CSF^{-/-} or G-CSFR^{-/-} mice. The authors are clearly aware of this option; in their discussion on p15/16 they provide a quite lengthy but not very convincing argumentation as to why these experiments were not done. Given that Csf3 and Csf3r knockout strains are commercially available, such experiments would be doable in a relatively short time frame.*

Indeed, the mice are commercially available (B6;129P2-Csf3tm1Ard/J, JAX no: 002398 and B6.129X1(Cg)-Csf3rtm1Link/J, JAX no: 017838), however they require cryo-recovery and are known to be poor breeders. Therefore, it would have required a significant amount of time to establish a colony needed to address the question. To make process faster, we were looking for the laboratories that breed those strains and initially we found a collaborator willing to help us with the experiments on G-CSF KO. However, the mice were breeding poorly and our collaborators were unable to provide the mice.

Given that we were not able to obtain sufficient numbers of mice to perform experiments proposed by reviewers, we decided to use the G-CSF blocking antibody to investigate the mechanism of CoPP action. Additionally, in another group of mice we used both anti-G-CSF and anti-IL-6 antibody, as we expected that the mechanism depends on synergistic effect of both cytokines.

We administered anti-G-CSF (each dose 12.5 μ g/mice, R&D, cat#MAB414, Monoclonal Rat IgG1 Clone # 67604, shown previously to neutralize G-CSF in vivo (Morris *et al*, 2015) or anti-G-CSF and anti-IL6 antibodies (each dose 130 μ g/mice, BioXcell, Monoclonal Rat IgG1 clone MP5-20F3 - (Tsukamoto *et al*, 2015) 1 hour before each dose of CoPP (total 5 doses during 5 consecutive days).

This strategy completely abolished G-CSF and IL-6 levels in plasma after 5 days of CoPP treatment, what demonstrates that our blocking approach was efficient (Fig 1, included in the manuscript as Fig 6A).

Fig. 1 G-CSF and IL-6 blocking.

Blocking of G-CSF significantly reduced the CoPP-induced mobilization of both granulocytes and HSC to the peripheral blood (Fig 2, included in the manuscript as Fig. 6B). This indicates that G-CSF is the major molecular mediator of the mobilization triggered by CoPP. Blocking both G-CSF and IL-6 did not provide any further reduction in number of mobilized granulocytes and HSCs, what suggests that IL-6 does not play a major role in CoPP-induced mobilization. This observation corresponds to the results of experiment where administration of G-CSF and IL-6 together did not provide any significant beneficial mobilizing effect (presented in the response to comments of Reviewer 2).

Fig. 2 The effect of G-CSF blocking.

Thus, we believe that we provided mechanistic evidence that CoPP mobilization is strongly dependent on G-CSF.

2. Another aspect that remains unanswered is the link with the CXCL12/CXCR4 axis, the other well-known pathway involved in stem cell mobilization/homing which is altered therapeutically with Plerixaflor to mobilize HSCs in G-CSF nonresponsive patients. Although CXCL12 (SDF1) does not appear to show up in the Luminex-based serum assays in CoPP-treated mice (Figure 1E), this does not exclude a possible role of the CXCL12/CXCR4 axis in CoPP-induced HSC mobilization. If this would be the case, CoPP might work as an alternative for Plerixaflor/AMD3100. Again, this possibility could easily be explored in commercially available Cxcl12 or Cxcr4 deficient mice.

We are grateful for the suggestion to check whether Cxcl12/Cxcr4 axis is involved in CoPP-induced mobilization. We investigated whether Cxcl12 and Cxcr4 expression is affected by the CoPP. We checked levels of Cxcl12 in the serum, Cxcr4 expression on mobilized HSPC in peripheral blood (LKS) and on HSC in the bone marrow, but we did not find any significant effect after 5 days of CoPP administration (**Fig 3**).

Fig. 3 CXCR4 expression on LKS cells in the peripheral blood and in the bone marrow.

Based on these results, and on the results showing that G-CSF mediates the major part of the observed mobilization, we did not proceed to evaluate effect of CoPP in Cxcl12 or Cxcr4 deficient mice.

Minor comments:

1. Introduction on CoPP-induced HO-1 enzyme activity is largely superfluous and can be condensed significantly.

We agree with the Reviewer and we shortened the introduction part about HO-1 activity.

2. Same applies to the extensive argumentation as to why G-CSF and IL-6 knockout strains were not used in the study (related to major comments above)

We excluded this argumentation in the revised version. Instead we shortly discuss the results of the experiment based on blocking of G-CSF.

3. A few typo's also in Figures, e.g., G-SCF in Figure 6D, need attention. We are grateful for bringing this to our attention. We revised the manuscript for the spelling mistakes.

Referee #2 (Remarks for Author): *This is an interesting manuscript by Szade and colleagues assessing the ability of CoPP to mobilize cells from the bone marrow, including HSPCs and granulocytes. This work has clinical implications in regard to patients with neutropenia, which leads to numerous detrimental consequences, and for patients in need of HSPC transplantation. It is also interesting that this process is independent of the Nrf1/HO-1 axis. Please see below additional comments / questions to be addressed by the authors. 1) In the experiments, the authors assess cell mobilization at different time points, for example in Figure 1 - 24 hours, and Figure 2 - 6 hours. Is there a rationale for varying the time point of analysis? The authors are also assessing different genetic background mice. Does the use of different mice reflect the variation in timing of harvest for analysis?*

The different time of analysis was not connected with the use of different mice strains. During the first experiment we did not expect that CoPP can induce any mobilization or G-CSF induction. Therefore, we used scheme of CoPP administration that was previously applied for pharmacological induction of HO-1 activity in mice (every second day for 5 days, samples collected after 24 hours). Those experiments led us to the observation that CoPP significantly induces G-CSF level and triggers mobilization. Our provisional results indicated that CoPP administration elevates G-CSF

levels within 3 hours. Thus, in all next experiments, where we wanted to compare the effects of recombinant G-CSF and CoPP, we applied the protocol that is commonly used for the mobilization studies with G-CSF (Levesque *et al*, 2004) (dosing each day for 5 days, collection at the last day of treatment).

[Unpublished data removed at the authors' request.]

Fig. 4 Cytokines induction after 1 dose of CoPP.

2) The authors feel that the CoPP effect for mobilization of HSPCs and granulocytes seems to be related to G-CSF and additional cytokines, such as IL-6. The authors discuss the challenges of confirming these findings using G-CSF or IL-6 deficient mice. However, to provide further insight into this hypothesis, have the authors tried to inject mice with a combination of G-CSF and cytokines (such as IL-6), to see if these mice respond in a manner more analogous to CoPP?

We are grateful for proposing the way to verify our hypothesis. We performed this experiment as proposed: we compared whether administration of G-CSF and IL-6 together can mimic beneficial mobilization effect obtained with CoPP. However, we did not observe any additional mobilization of granulocytes after administration of both G-CSF and IL-6 compared to G-CSF alone (**Fig 5**).

Fig. 5 G-CSF and IL-6 co-administration effect on mobilization.

Additionally, we blocked G-CSF or both G-CSF and IL-6 with neutralizing antibodies during the CoPP mobilization (explained above in details in response to remark 1 of Reviewer 1). Consistently, while blocking G-CSF significantly abolished mobilization of both granulocytes and HSC, additional IL-6 blocking did not reveal any effect. (**Fig. 2**, included in the manuscript as Fig. 6B). Therefore, we concluded that mechanism of CoPP-induced mobilization is dependent on G-CSF, but not on IL-6.

3) The authors demonstrate CoPP induces mobilization of HSPCs that upon transplantation into irradiated mice have faster reconstitution and higher blood chimerism compared with cells mobilized by G-CSF. Do the authors know the mechanism for this improved response? Did the authors assess the cytokine panel (Figure 1) in irradiated mice that received HSPCs from CoPP and G-CSF mobilized cells? Can the authors give additional insight into this beneficial response?

After the transplantation we collected the blood samples from living mice by submandibular bleeding. This allowed for collection of around 50 μ l of blood, which we used for total blood cell count and flow cytometry analysis, so technically, we did not have enough material for assessing the cytokine concentration.

We proposed that the mechanism of the faster reconstitution and higher blood chimerism is

connected with number of the mobilized cells. While the level of the long-term chimerism in transplantation assays is linked to long-term HSCs, the first blood cells are produced by short-lived progenitors (Purton & Scadden, 2007). During our analysis we used detailed antigen profiling that allowed us to distinguish the long-term HSC and short-lived progenitors. While, the experiments presented in the initial manuscript already showed more progenitors and LT-HSC, we performed additional analysis of new sets of experiments.

Consistently, we observed that both the number of long-term HSC and short term progenitors (granulocyte-monocyte progenitors GMP - L-K+S-CD150-CD48^{high}CD34⁺, and early megakaryocyte-erythroid progenitors eMEP - L-K+S-CD150^{mid}CD48^{mid}CD34⁺) are significantly more abundant after CoPP-induced mobilization than after administration of rhG-CSF. This likely explains that CoPP provides both faster reconstitution as well as higher level of long-term chimerism. We clarified this in the discussion.

Fig. 6 Frequency of lineage-committed progenitors in the peripheral blood.

2nd Editorial Decision

25 September 2019

Thank you for the submission of your revised manuscript to EMBO Molecular Medicine. We have now received the enclosed reports from the referees that were asked to re-assess it. As you will see the reviewers are now globally supportive and I am pleased to inform you that we will be able to accept your manuscript pending minor editorial amendments.

Please submit your revised manuscript within two weeks. I look forward to seeing a revised form of your manuscript as soon as possible.

***** Reviewer's comments *****

Referee #1 (Remarks for Author):

The authors have adequately dealt with my comments, added new data and and revised the Ms appropriately.

Referee #2 (Remarks for Author):

The authors have answered all of my questions.

2nd Revision - authors' response

9 October 2019

Authors made the requested editorial changes.

YOU MUST COMPLETE ALL CELLS WITH A PINK BACKGROUND ↓
PLEASE NOTE THAT THIS CHECKLIST WILL BE PUBLISHED ALONGSIDE YOUR PAPER

Corresponding Author Name: Agata Szade

Manuscript Number: EMBO-2018-0971